# Review on Torque Distribution Scheme of Four-Wheel In-Wheel Motor Electric Vehicle

**Shuwen He** [1] , **Xiaobin Fan** [1,*] , **Quanwei Wang** [1], **Xinbo Chen** [2] **and Shuaiwei Zhu** [1]

1    School of Mechanical and Power Engineering, Henan Polytechnic University, Jiaozuo 454000, China; hsw990411@163.com (S.H.); wqw17719986983@163.com (Q.W.); zhshuaiwei@163.com (S.Z.)
2    China North Vehicle Research Institute, Beijing 100072, China; chenxb_2008@163.com
\*    Correspondence: fanxiaobin@hpu.edu.cn; Tel.: +86-150-3651-2985

**Abstract:** In-wheel motor electric vehicles have the advantages of independently controllable four-wheel torque, high energy utilization rate, and fast motor response speed, which greatly reduces the curb weight of the vehicle and simplifies the structure of the vehicle, making it an expert at home and abroad research hotspots. However, the in-wheel motor independently drives the electric vehicle. The in-wheel motor directly drives the vehicle, and the motion states of each wheel are independent of each other; that is, each wheel can be independently driven by wire control, which puts forward higher requirements for the torque distribution control of the entire vehicle. Starting from the driving form of the car, this paper focuses on the design of the torque distribution scheme of the in-wheel motor by experts and scholars in the past, such as the use of genetic algorithm, BP neural network, particle swarm algorithm, and fuzzy control algorithm to distribute the torque of the in-wheel motor, and the research on vehicle economy and stability under torque distribution optimization is reviewed. The future development direction of in-wheel motor torque distribution is prospected.

**Keywords:** four-wheel in-wheel motor vehicle; in-wheel motor; vehicle dynamics modeling; intelligent control algorithm; torque distribution optimization



## 1. Introduction

In recent years, the energy crisis and a series of ecological problems brought about by environmental pollution are further promoting the development of automobiles from traditional fuel vehicles to electric vehicles with the concept of energy saving and environmental protection. Four-wheel independent drive electric vehicles based on in-wheel motors are one of the more promising ones. The four-wheel independent drive electric vehicle based on the in-wheel motor, the motion state of each wheel can be independent of each other, and there is no rigid mechanical connection between them, which greatly improves the transmission efficiency, reduces the curb weight of the whole vehicle, simplifies the vehicle structure, and is conducive to increasing the electric vehicles' range, so in-wheel motor drive is considered the final form of drive for electric vehicles. The four-wheel drive torque of the four-wheel independent drive in-wheel motor electric vehicle is independently controllable, which is convenient for integrated control by using wire-controlled technology and wire-controlled steering systems. Compared with traditional internal combustion engine vehicles, it has more controllable degrees of freedom. It is an ideal carrier for a new generation of vehicle control technology and for exploring the optimal dynamic performance of vehicles [1]. Additionally, the torque distribution problem of the four-wheel in-wheel drive electric vehicle has a decisive influence on the driving economy and stability. Based on the reasonable distribution of the in-wheel motor torque, the research methods can be divided into handling stability control and energy management control [2,3]. In the aspect of handling stability control, such as the Electronic Stability Control System (ESP) [4,5] and Front Wheel Active Steering System (AFS) [6,7], etc., the economic aspects include the braking torque of electric vehicles. The rational distribution of torque in two

parts [8], energy-saving distribution and driving torque energy-saving distribution, is the research focus, so it is of great value and significance to study the in-wheel motor torque distribution scheme. However, the current research on the torque distribution of in-wheel motors is not in depth and comprehensive enough. Most of them are under ideal circumstances, ignoring the constraints to optimize the torque distribution of a single performance parameter. Therefore, it is necessary to consider the actual constraints in the future. Comprehensive research to improve the accuracy and practicability of the in-wheel motor torque distribution scheme.

This paper firstly introduces the driving form of electric vehicle, and briefly explains the structure and principle of in-wheel motor driving electric vehicle. Then, the in-wheel motor and vehicle dynamics modeling of some experts and scholars are introduced, and then the current in-wheel motor torque distribution schemes are introduced from the aspects of improving stability and energy efficiency, and their advantages and disadvantages are analyzed. Finally, the above the schemes are compared and summarized, and the research direction and development ideas of the in-wheel motor torque distribution strategy are prospected and summarized.

## 2. The Driving Form of the Motor Vehicle

Electric vehicle drive is divided into traditional centralized drive structure type and distributed drive electric vehicle structure type [9,10]. The structure of traditional centralized drive is similar to that of fuel vehicles, so this paper mainly discusses the drive structure of distributed drive electric vehicles. Distributed drive is divided into wheel motor drive and in-wheel motor drive.

### 2.1. Wheel Side Motor Drive

Both the wheel side motor and the hub motor use the motor to drive the wheel hub. The wheel side motor is separated from the wheel hub and the motor, and the motor is installed next to the wheel to work, as shown Figure 1 [11]. The electric vehicle motor installed on the body in the form of wheel-side motor drive has a great influence on the overall layout of the vehicle, especially in the case of rear axle drive. The universal joint transmission of the drive shaft also has certain limitations. Using two motors and two controllers, in order to meet the motion coordination of each wheel, the requirements for the synchronous coordinated control of the two motors are high, which increases the design difficulty of the electronic control system. The distributed installation arrangement of the electric motor also brings many technical problems such as structural arrangement, thermal management, electromagnetic compatibility, and vibration control. In terms of motor selection, at present, the mainstream types of drive motors include AC asynchronous induction motors, switched reluctance motors, and permanent magnet synchronous motors. Compared with AC asynchronous induction motors or switched reluctance motors, permanent magnet synchronous motors have the advantages of high efficiency, wide operating range, large power factor, and high-power density, and become the first choice for wheel-side direct drive motors [11].

### 2.2. In-Wheel Motor Drive

The in-wheel motor is a structure that directly integrates the motor with the wheel, as shown in Figures 2 and 3 [12]. The in-wheel motor drive system is mainly divided into two structural forms according to the rotor form of the motor: inner rotor type and outer rotor type. The inner-rotor hub motor adopts a high-speed inner-rotor motor, equipped with a reducer with a fixed transmission ratio, and the speed of the motor is usually as high as 10,000 r/min. The outer rotor hub motor adopts a low-speed outer rotor motor without a reduction device. The outer rotor of the motor is fixed or integrated with the rim of the wheel. The speed of the wheel is the same as that of the motor, and the maximum speed of the motor is between 1000 and 1500 r/min. The inner rotor type in-wheel motor has the advantages of high specific power, light weight, small volume, low noise, and

low cost. The disadvantage is that a deceleration device must be used, which reduces the efficiency and increases the unsprang mass. The maximum speed of the motor is limited by factors such as coil loss, friction loss, and the bearing capacity of the transmission mechanism. The advantages of the outer rotor hub motor are simple structure, small axial size, torque control in a wide speed range, fast response, no reduction mechanism, and high efficiency. The disadvantage is that to obtain a larger torque, the volume and mass of the motor must be increased, so its cost is high. Both of these structures have applications in current electric vehicles, but with the advent of compact planetary gear transmissions, high-speed inner-rotor drive systems are more competitive than low-speed outer-rotor drives in terms of power density. Different from the traditional drive form, the in-wheel motor drive car has opened up a new world for the research and development of the car with a new structure. Since the torque of each wheel is independently controllable, this new type of drive enables advanced dynamic control. In addition, the removal of powertrain and transmission system gives car designers more freedom in optimizing the layout of the vehicle. With the rapid development of motor and battery technology, batteries and motors with smaller volume and higher energy density will bring better opportunities for the research and development of this type of electric vehicle [13]. By then, the usability and comfort of the vehicle will be improved with a big boost.

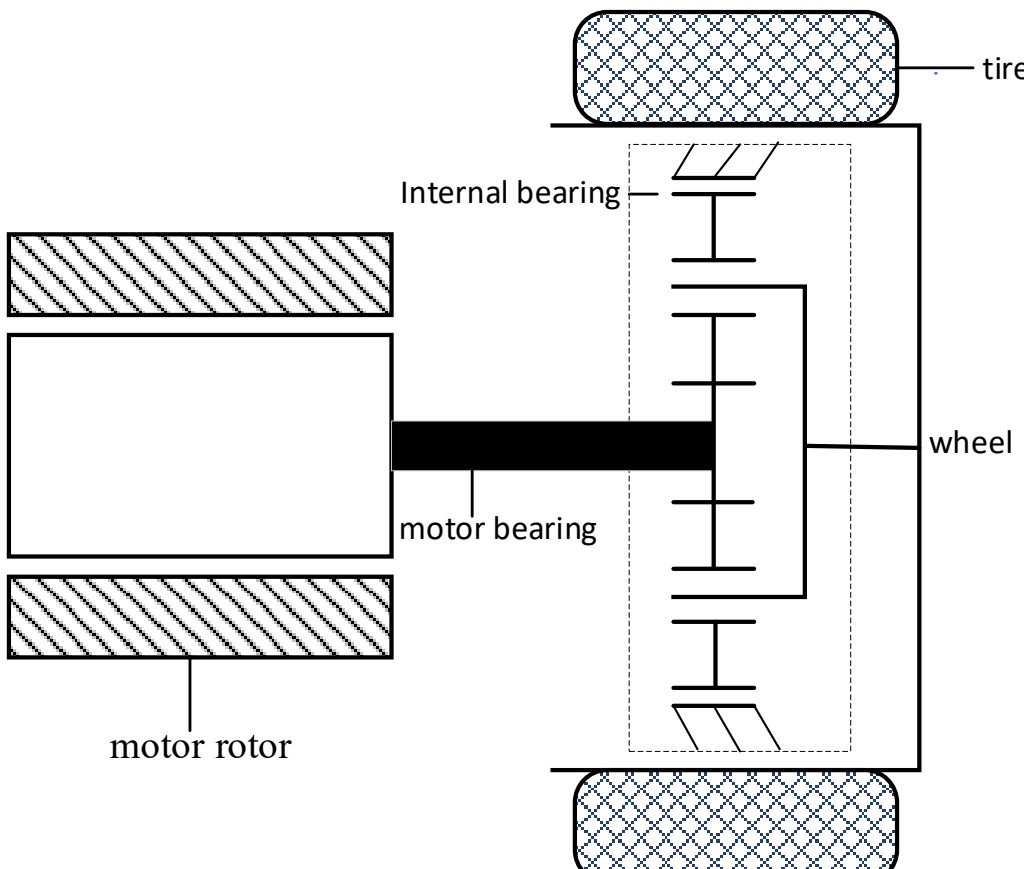

**Figure 1.** Structure characteristic diagram of wheel side drive.

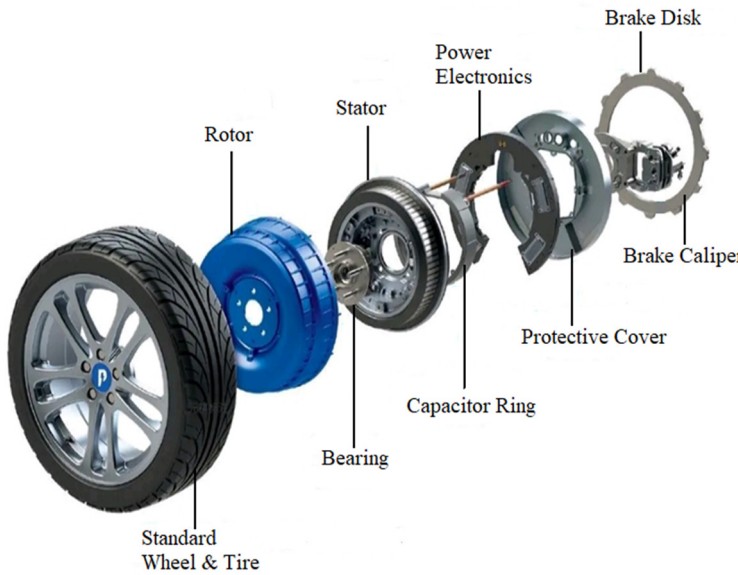

**Figure 2.** Structure diagram of in-wheel motor.

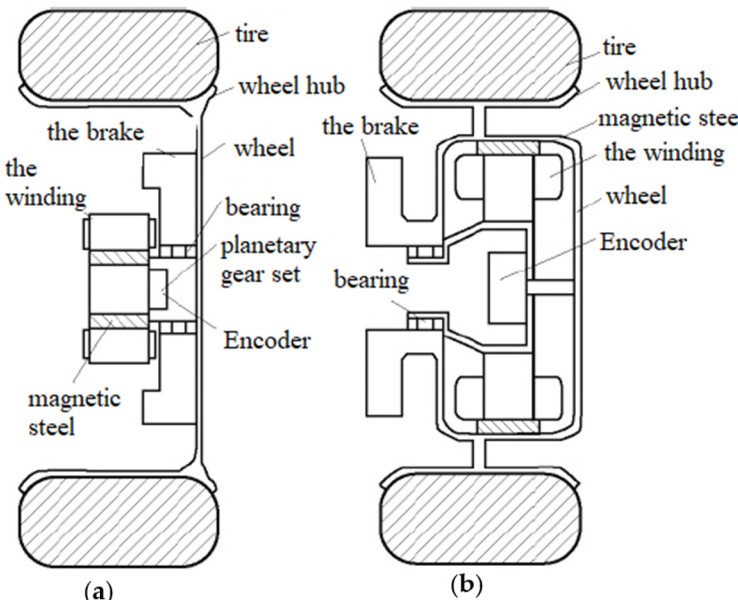

**Figure 3.** Schematic diagram of in-wheel motor structure (**a**) Deceleration drive (**b**) Direct drive.

## 3. In-Wheel Motor Drive Torque Distribution Scheme

### 3.1. In-Wheel Motor Drive Torque Distribution Method

Compared with the centralized drive electric vehicle, the torque of the four wheels of the in-wheel motor-driven electric vehicle can be independently controlled, and the torque can be distributed according to any proportion within its capability range, so as to improve the vehicle handling stability and smoothness. However, the dynamic problems caused by the driving mode and structural form of electric vehicles, especially the overdrive characteristics caused by distributed driving, have brought opportunities and challenges to the vehicle dynamics stability control. Therefore, scholars at home and abroad have adopted various methods to reasonably distribute the torque of the in-wheel motor, so that it can better meet the operational stability and smoothness of the vehicle. In modern research, the reasonable distribution of the expected performance and torque of the car is usually divided into upper and lower layers for hierarchical control. For example, according to the definite relationship between the yaw rate and the wheel torque, the

wheel torque can be directly calculated from the expected value, or proportional wheel torque distribution according to the vertical load of the tire. The following mainly analyzes the methods adopted by domestic and foreign experts and scholars in the distribution of driving torque of in-wheel motors from the aspects of stability and energy efficiency.

### 3.1.1. In-Wheel Motor Drive Torque Distribution Method to Improve Stability

(1)  In-wheel motor drive torque distribution method based on vehicle dynamics model to improve stability.

a.  Torque distribution considering road adhesion

Huang [14] adopted the torque vector distribution strategy based on the premise of the optimal slip rate and realized the given desired direct yaw moment through the braking wheel and the driving wheel at the same time; that is, differential driving and differential braking were adopted. The torque vector distribution control method combined with driving can reduce the wheel torque output, improve the utilization rate of road adhesion limit, and then expand the stability margin of the vehicle. The proposed regularized torque distribution control strategy can reliably achieve the desired direct yaw moment and improve the safety of vehicles on low-adhesion roads. Based on the established linearized Burckhardt tire model, the road surface utilization adhesion coefficient can be accurately identified. Through this control method can effectively improve the lateral stability of the vehicle, it does not consider the uncertainty of the motor output torque, so the robustness of torque distribution needs to be further improved.

Zhao [15] controlled the distribution of driving force based on equal torque and equal power, respectively, calculated the deviation of the adjusted amount of each driving wheel under the current driving state, and converted this deviation into a current deviation, which was sent to the drive system controller to realize the distribution control of the driving torque. This method does not control the absolute value of each drive motor but controls the relative quantity relative to the whole vehicle; that is, the feedback controller is used to control the drive force of each drive system at the system state point determined by the superior drive torque distribution strategy. However, this method only calculates the driving current of each wheel according to a certain law, without considering the working state of each wheel and taking control measures, so Zhao [15] further proposed the driving force distribution based on the composite slip rate. Taking the lateral force of each wheel as a constraint, the study focused on optimizing the longitudinal driving force of each wheel, so as to obtain better longitudinal driving performance and better lateral stability. In this way, each wheel operates in coordination with each other, thereby driving the vehicle to run smoothly. The proposed method is based on equal torque and equal power to control the driving force distribution, the control program is easy to implement, and the calculation method is also simple and easy to implement, and then based on this, a compound slip rate control strategy is proposed to improve the shortcomings of the previous scheme, so that the coordinated operation of the drive system is more optimized.

Xiao [16] considered that the hierarchical scheme can be easier to implement diagnosis and fault-tolerant processing [17,18], so a hierarchical architecture was used to design the DYC system, by considering the longitudinal, lateral, and yaw directions of the vehicle body. Based on the stability criterion, a sliding mode surface switching mechanism is established to calculate the total motion force and total torque. Then, considering the road adhesion and motor peak torque constraints, the square sum of tire utilization with weight coefficient is selected as the stability. According to the optimization objective of the performance control, the wheel torque is optimally distributed, and the overall frame diagram is shown in Figure 4 [16].

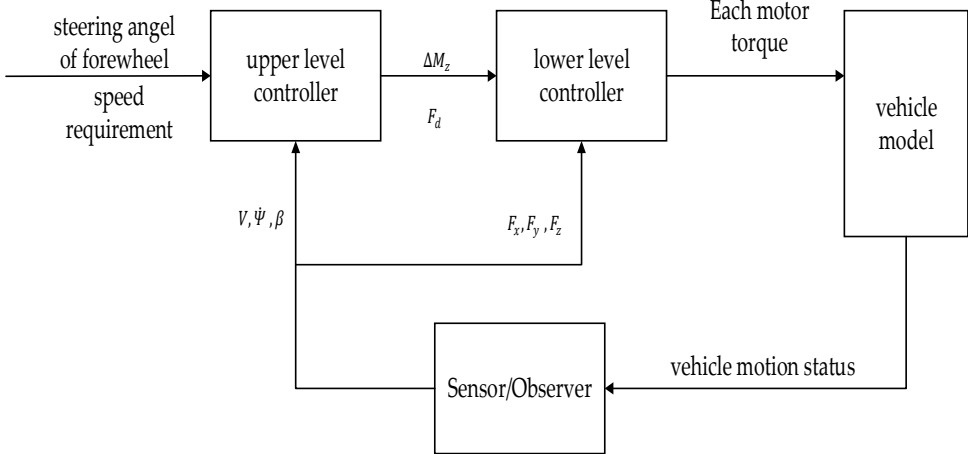

**Figure 4.** Frame design for vehicle yaw moment control.

In the designed lower-level controller of torque distribution, firstly, the tire load rate is the primary consideration index [19], and the minimum sum of squares of all wheel load rates is taken as the optimization objective to design the lower-level controller. Under different driving conditions, the vehicle can timely distribute the motor torque according to the vertical load of each wheel, and on the premise of maintaining the stability of the vehicle, it can also avoid the vehicle from slipping. Then, considering that the road adhesion conditions are relatively poor, the algorithm that minimizes the sum of the squares of the load rate as the optimization objective has no solution, so the ratio of the loads on each axle is used to distribute the torque of each wheel, and the weight coefficient is calculated, so that the optimized torque distribution results can be obtained after the solution.

b.  Torque distribution considering tire force factors

Ling [20] proposed an equal-proportional torque distribution and an optimal distribution method based on the weight of the center of mass slip angle and calculated the weight coefficients of the front and rear axles [21]. The torque of the in-wheel motor is distributed when the parameters are affected, and the optimization goal of the torque optimization distribution strategy is to minimize the sum of the squares of the utilization rates of the four tires. In this way, each wheel can play its role, allowing the wheel with greater adhesion to have a larger driving force or braking force, and improving the stability margin of the in-wheel motor-driven electric vehicle. The steering performance of the vehicle is guaranteed, and the driving stability of the vehicle is improved.

Guo [22] considered the robustness of the previous stability control strategies without considering the tire force saturation factor, slip rate, and adequacy of the control system. According to the characteristic that the wheel torque can be distributed independently, a stepwise torque distribution strategy under emergency conditions is proposed. The algorithm consists of three levels of controllers, namely, the upper controller uses the phase plane method [23] to determine whether the car is in a stable state, the intermediate controller calculates the required traction and yaw moment, and then it converts these virtual signals into the underlying controller of the actual executor command [24]; the principle is shown in Figure 5 [22].

Considering the three factors of slip rate constraint, in-wheel motor constraint, and tire force constraint in torque distribution, the cost function of distribution error and adhesion utilization rate is designed. Finally, the problem is transformed into a canonical least squares problem, which can be solved using the active set method. The final simulation results show that the algorithm can improve the handling stability of the four-wheel drive electric vehicle in emergency conditions.

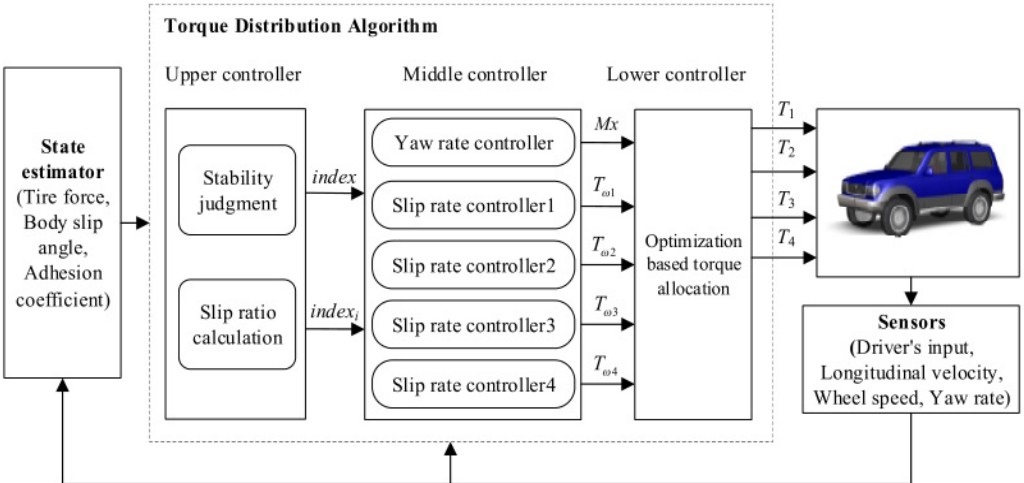

**Figure 5.** Schematic diagram of the torque distribution algorithm.

c.    Torque distribution considering driver factors

Tahami [25] proposed a power augmentation system suitable for front-wheel-steering all-wheel-drive electric vehicles. The study utilizes direct yaw rate feedback to help drivers correct road deviations and improve vehicle cornering and acceleration. Considering that the torque distribution of the power system is composed of multiple nonlinear problems, the central fuzzy controller [26] is used to control the yaw angular velocity of the vehicle, and then four other fuzzy controllers are set up to control the yaw rate caused by the yaw controller, respectively, so that the tire force reaches the saturated torque and prevents the driving wheel from slipping. When the neural network algorithm model used in the experiment is used to simulate the generated training examples for back-propagation training of the neural network, the amount of training is usually large, and the pre-experiment takes a long time.

Mutoh [27] studied the torque distribution of an in-wheel trolley with independent front and rear wheels driving and braking under low μ-level curves and braking conditions. From the safety point of view, considering that the current anti-lock braking system (ABS) cannot work normally when the brake pedal pressure is small [28], a strategy of reasonable torque distribution is adopted to ensure the safety of the car. Using the linear control system design theory based on the continuous control rule, the regulator of the torque controller is designed, so that the torque controller can maintain stability against the transient change of the compensation torque reference point and the torque control of the motor can be adjusted in the torque-μ curve in the stable region. It ensures the safety of the car on the road with low friction coefficient.

Using Terminal Sliding Mode Technology (TSMC) [29], Song [30] proposed a torque distribution method for single-wheel drive vehicles by interpreting driver commands to track the desired vehicle motion. By considering the nonlinear constraints of the tire sticking limit, a simple and effective assignment strategy is introduced to efficiently distribute the motion control to each wheel. The motion control force and the controlled variable are transformed into a functional relationship, and the motion control work generated by the TSMC needs to be reasonably distributed to the motor that drives the four wheels. This enables the driver to maneuver the vehicle more easily without steering force compensation for the non-linearity of the vehicle's yaw response. Finally, the terminal sliding mode control (TSMCR) has fast, limited time convergence and high steady-state accuracy through simulation, so it has better control effect than the synovial film control (SMCR) [31].

(2)    In-wheel motor drive torque distribution method based on algorithm to improve stability.

a.    Calculate the torque allocation of the optimal yaw moment algorithm

Jin [32] started from the constraint conditions to study the torque distribution. These include the saturation constraints and oblique constraints of the motor, the characteristics of the tires attached to the road surface, the road surface adhesion, the vertical load of each wheel and other factors, and the driving anti-skid constraints on low-adhesion roads or when the driving torque of the in-wheel motor is too large. Through the error back propagation algorithm based on BP neural network and the fastest descent method [33], the author adjusts the yaw moment algorithm of PID to calculate the optimal yaw moment, then considers various constraints, and then calculates the driving torque of each wheel allocate solve. This method simulates the driving environment of the car more realistically by considering various constraints and can make good use of the adhesion conditions of the road surface and prolong the life of the motor when the motor torque is distributed. However, only a part of the constraints is considered, and subsequent research can carry out further constraints on this basis.

Zhang [34] asked professional drivers to conduct experiments from the perspective of drivers, and obtained the expected torque of professional drivers, taking into account the lateral and longitudinal stability for optimal torque distribution, as shown in Figure 6 [34]. Then, based on the idea of hierarchical control, the Adaptive Second Order Sliding Mode (ASOSM) [35] algorithm is used to calculate the desired torque for optimal distribution. This method reflects the driver's dominance and broadens the research field of optimal torque distribution. The adaptive second-order sliding mode control also overcomes the chattering problem of traditional sliding mode and improves the robustness of the control system to model errors and parameter uncertainties. However, different drivers' driving styles and driving habits are not considered, and experimental research can be carried out in this area in the future.

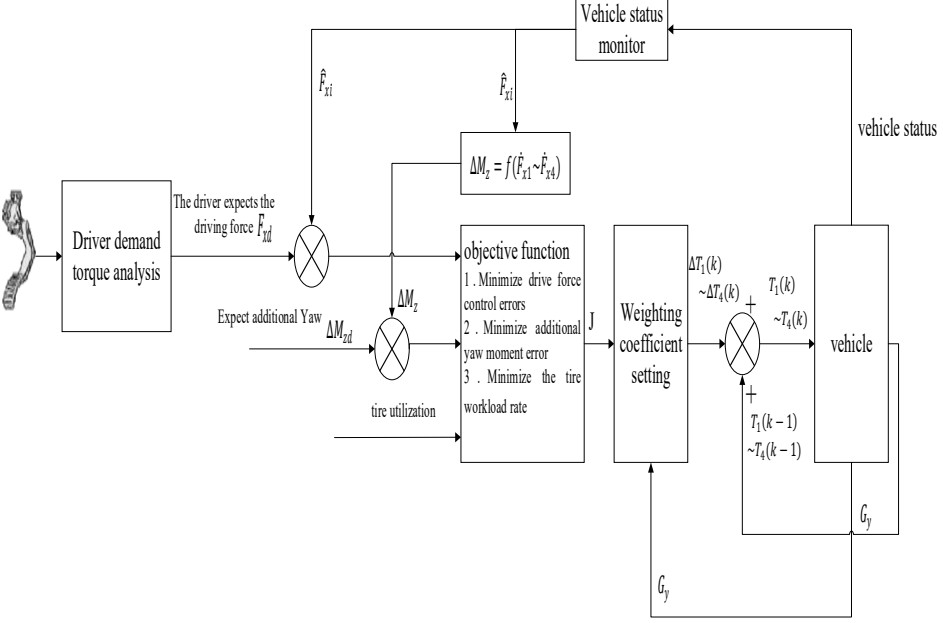

**Figure 6.** Optimal torque distribution strategy architecture.

b.   Torque distribution of quadratic programming algorithms

Shuai [36] considered factors such as communication delay and interruption in distributed control systems and designed a four-wheel torque optimal distribution algorithm with communication fault tolerance. Based on the vehicle four-wheel rigid body model [37] and the Magic Formula Tire Model (MFTM) [38], using the control distribution theory of the overdrive system, the torque optimization distribution problem is transformed into a quadratic programming problem [39], and through the reasonable design of the coefficient matrix in the quadratic programming problem, the reasonable optimization of the four-wheel attachment utilization is realized under the premise of satisfying the upper-level

control commands. Then, considering the random network message transmission delay and continuous network communication interruption failure, a set of fault diagnosis and fault tolerance mechanism is established, so that the vehicle controller and motor controller can work in fault-tolerant mode when network problems occur, and the network can be switched to the normal mode when the network is normal, so that the torque can be optimally distributed. However, the author also pointed out that the delay signal in the feedback channel was not considered, the long-term delay was not experimentally studied, and separate experimental verifications are required in the future.

Yan [40] considered that the wheels on both sides of the in-wheel motor-driven tram could not naturally form a differential effect during the steering process, and the active action of the electronic control system was needed to improve the difference in the distance between the inner and outer wheels during the vehicle steering process to prevent tire wear and improve vehicle steering maneuverability [41]. Therefore, Yan Chunhui et al. conducted experimental research on the torque control of low-speed steering and differential speed, taking into account the auxiliary steering and the driving force control of high-speed steering aiming at stability. In the case of low-speed steering, the author adopts Active Disturbance Rejection Control (ADRC) [42] to effectively solve the contradiction between overshoot and rapidity of the classical controller and makes the parameters of feedback gain and error differential feedback gain convenient. Tuning can enhance the robustness of the control system and improve the accuracy of the calculation. Under high-speed steering, the quadratic programming algorithm [43] is used first for the purpose of stability, as shown in Equation (1).

$$
\begin{aligned}
min \tfrac{1}{2} x^T H x + C^T x \\
\text{S.T. } A_{eq} x = b_{eq} \\
A x = b
\end{aligned}
\tag{1}
$$

where $H$ is the Hessian matrix. According to the yaw rate of the vehicle and the side-slip angle of the center of mass, the motor torque is coordinated and distributed to ensure the stability of the vehicle steering. However, the influence of vehicle roll on the distribution of driving force was not considered in the experiment, but the effect of vehicle suspension was simplified. Future research needs to be further experimentally verified in this regard.

Li [44] took the friction circle limit, the proportion of longitudinal force and lateral force, the motor torque limit, and the motor failure limit as constraints; the vehicle sensor system collected the real-time longitudinal speed $V_x$ of the vehicle and the real-time rotation speed of each tire $\omega_i$, the real-time driving torque $T_i$ of each in-wheel motor, the longitudinal acceleration $a_x$, and the lateral acceleration $a_y$, and then the real-time adhesion capacity of each tire is calculated through the tire friction circle estimation. At the same time, the real-time load $F_{zi}$ of each tire is calculated through the three-degrees-of-freedom vehicle model. Substitute the obtained results into the objective function and constraint conditions of objective optimization as known conditions, and then determine the longitudinal force weight factor $C_x$ and lateral force weight factor $C_y$ in the objective function through the weight coefficient calculation module; finally, the objective optimization algorithm can be calculated. The optimal tire longitudinal force and lateral force are finally converted into the driving torque and steering wheel angle correction value of each tire by the parameter correction module, as shown in Figure 7 [44]. Then, considering the failure of the in-wheel motor, the failure parameter $\varepsilon_i$ is used to verify the constraints and the objective function. The final objective function is a concave function, and the optimization problem is finally transformed into a concave function extremum problem. The algorithm design of this study only considers the steering of the front wheel and ignores the influence of the lateral force of the rear wheel on the vehicle, which needs to be further optimized in the future.

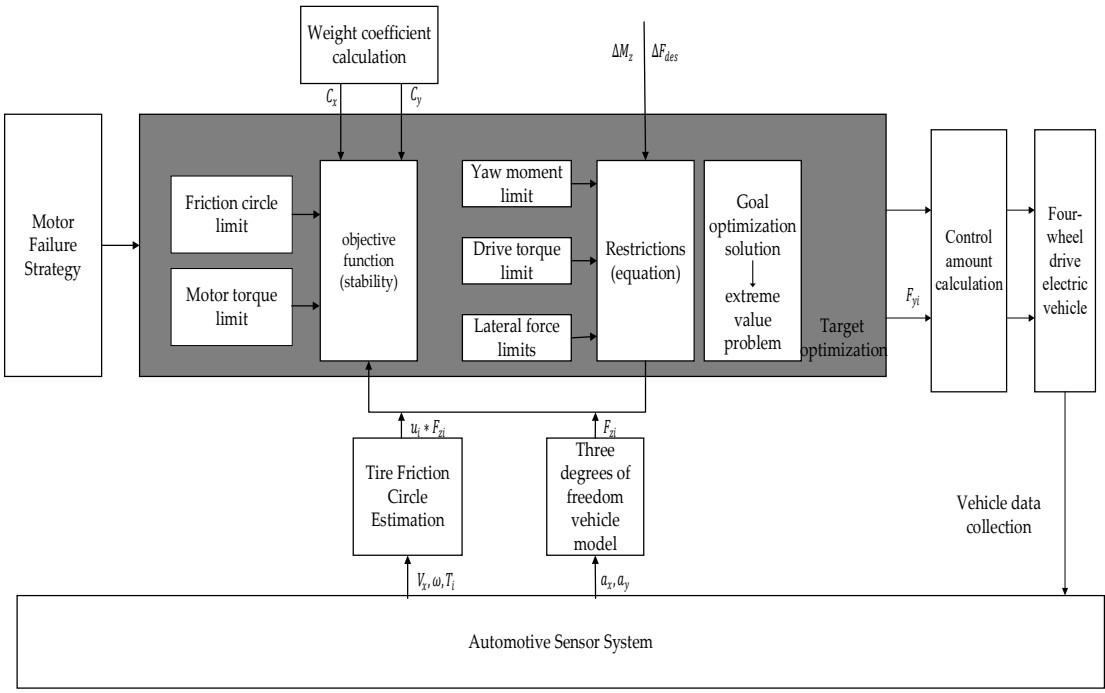

**Figure 7.** Overall structure of torque distribution strategy for four-wheel in-wheel motor independently driven electric vehicle.

Zhai [45] are based on hierarchical control rules, the upper layer consists of a speed tracking controller, a yaw torque controller, and four-wheel slip controllers, which are used to calculate the expected value of the traction force, the expected value of the yaw torque, and the expected value of the four wheels' net torque input to the motor. Using the average torque distribution strategy, the dynamic tire load-based torque distribution strategy, and the optimal torque distribution strategy based on the minimum objective function, the torque optimization distribution problem is transformed into a quadratic programming problem to control the motor drive torque or regenerative braking torque, thereby improving vehicle stability.

### 3.1.2. In-Wheel Motor Drive Torque Distribution Method to Improve Energy Efficiency

(1) In-wheel motor drive torque distribution method based on vehicle dynamics model to improve energy efficiency.

a. Torque distribution considering motor energy efficiency optimization

Gu [46] studied the motor efficiency MAP, considering that the total power loss of all motors can be minimized under the condition of the same total output power, and the energy efficiency of the electric drive system can be maximized, so the power loss of the motor can be maximized. The torque distribution optimization problem is analyzed from the perspective. Further theoretical analysis shows that the total torque demand should be evenly distributed to each in-wheel motor to maximize the overall energy efficiency of the electric drive system, regardless of the size of the total torque demand. Through further experimental research, it was confirmed that the proposed scheme of torque equal distribution among four wheels is superior to the previous two-wheel/four-wheel switching distribution scheme [47,48]. However, this scheme has only been tested under specific motor and algorithm conditions, and whether it is applicable to other motors and algorithms needs further experimental verification.

Wu [49] conducted research based on the analysis model of permanent magnet synchronous motor (PMSM) loss to study the optimal torque distribution rate with minimum system loss. They analyzed the case where the motor parameter values of the front wheel

and the rear wheel are the same and the case where the motor parameter values are different. The analysis and experiment show that if the motor loss is a convex function of the electromagnetic torque under the same conditions, the uniform torque distribution between the front and rear wheels will minimize the system loss, and the optimal torque distribution coefficient depends on the motor parameters in different situations. Wu Dongmei et al. also incorporated no-load losses into the system efficiency data and used numerical optimization methods to calculate the optimal distribution coefficient.

Yuan [50] proposed a torque distribution scheme based on a permanent magnet motor loss model to optimize motor efficiency. The relationship between the motor efficiency and the torque distribution ratio at a given speed is derived, and it is proposed that at high torque, the torque should be equally distributed between the two motors to achieve maximum efficiency (or minimum loss). When the vehicle is under low torque conditions, determine the torque boundaries for dual-motor operation and single-motor operation with equal torque sharing, and use the clutch to combine the motor efficiencies and the reasonable distribution of work with a single motor or dual motors to maximize efficiency. Finally, the simulation verification shows that the motor losses are greatly reduced, and the energy efficiency is also improved compared with the New European Driving Cycle (NEDC).

b.    Torque distribution considering tire slip energy loss

Ren [51] first proposed the torque optimization control for straight driving conditions by combining the power system energy consumption index and the vehicle road adhesion utilization index in the performance function. Under the condition that the road adhesion cannot be accurately obtained in real time, it is an "Energy-and-Adhesion" (EAA) joint optimization problem to synergistically optimize the energy efficiency of the power system and the utilization of road adhesion. In the calculation, a cubic polynomial is used in combination with the least squares method to fit the function of the output torque. At the same time, it is compared with the quadratic polynomial fitting method [52] and the piecewise linearization fitting method [53], and the results show that the cubic the fitting effect of polynomial is obviously better than the above two. Under the steering condition, he proposed a torque distribution method that coordinated the direct yaw moment control action and the steering characteristic improvement requirement. The hierarchical control architecture was used to realize the synergistic optimization of the energy efficiency and steering characteristics of the chassis system under the steering condition [54]. The upper layer adopts the model predictive control algorithm considering the execution constraints, and the lower layer takes the tire load rate and the energy consumption of the electric drive system as the optimization targets under the constraints of the target yaw moment and total longitudinal moment demand and deduces the global optimum according to the minimum value principle four-wheel torque distribution method. Finally, the simulation analysis proves the feasibility of the method.

Yu [55] proposed a torque distribution method that minimizes the loss of the entire system under the condition of satisfying the load torque demand based on the modular permanent magnet in-wheel motor model. Then, the formula method loss minimum control and the search method loss minimum control are respectively performed. The two are compared, and finally the search method with the least loss control is selected. Then, the loss-minimizing control and the conventional control are compared under low-speed and high-torque conditions, rated conditions, and high-speed conditions. It is concluded that the loss-minimizing control has a higher efficiency improvement in the high-speed range at rated torque and the light-load range at rated speed.

(2)    In-wheel motor drive torque distribution method based on algorithm to improve energy efficiency.

a.    Torque Distribution of Multi-motor Cooperative Control Algorithm

Meng [56] adopted the fast search method (the golden section method), used the interval elimination method to narrow the search interval, iteratively calculated to further

narrow the search range, and finally obtained the approximate solution of the corresponding value of the extreme point, as shown in Figure 8 [56].

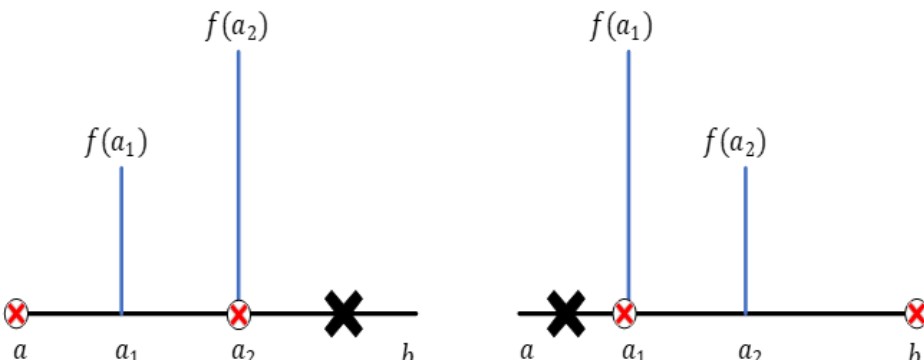

**Figure 8.** Schematic diagram of fast search method interval elimination.

The search process is divided into the following: one is to determine the initial search range and the accuracy of convergence; the second is to calculate the insertion point and function value according to the formula; the third is to use the principle of interval elimination to narrow the search interval; finally, check whether the interval satisfies the condition of convergence, and the process is shown in Figure 9 [56].

Through this process, the total required torque is calculated according to the distribution coefficient k during torque distribution during torque distribution control, and then the front and rear wheels are distributed. When assigning, Meng Bin et al. considered the existence of variable factors in the process of steering and driving, and also carried out fuzzy control from a nonlinear perspective [57,58]. The proposed fast search method can make the driving torque distribution ratio of the front and rear axles controlled in the working area with the optimal driving efficiency of the system without being constrained by the parameters of the motor and other components. However, the algorithm will have fluctuation problems, which need to be solved by further calibration, and the fuzzy algorithm will also have relatively large errors, which all need to be solved [59].

Sun [60] obtained the feasibility of applying different driving torques to each wheel through the theoretical analysis and experimental verification and other methods, so as to reduce the vehicle turning energy consumption. It is pointed out that the essence of the torque distribution control of the minimum turning energy consumption is to independently control the output torque of the inner and outer wheel hub motors of the vehicle and output the required torque according to the control target, so the key point of this idea is to coordinately control the torque of multiple motors and use an algorithm to find the optimal solution [61]. Sun [60] chose to use the genetic algorithm [62,63] to solve the problem, and the mathematical model can be expressed as [64]:

$$SGA = (C, E, P_0, N, \Phi, \Gamma, \Psi, T) \tag{2}$$

Among them, $C$ is the coding method of the individual, $E$ is the individual fitness evaluation function, $P_0$ is the initial population, $N$ is the population size, $\Phi$ is the selection operator, $\Gamma$ is the crossover operator, $\Psi$ is the mutation operator, and $T$ is the termination of the genetic operation condition.

The solving steps include the following: first analyze the problem, obtain the constraints and control variables, establish the algorithm model, determine the chromosome coding form, determine the decoding method, determine the quantitative average index of the individual's fitness, design the genetic operator, and finally determine the genetic algorithm. The relevant operating parameters are shown in Figure 10 [60].

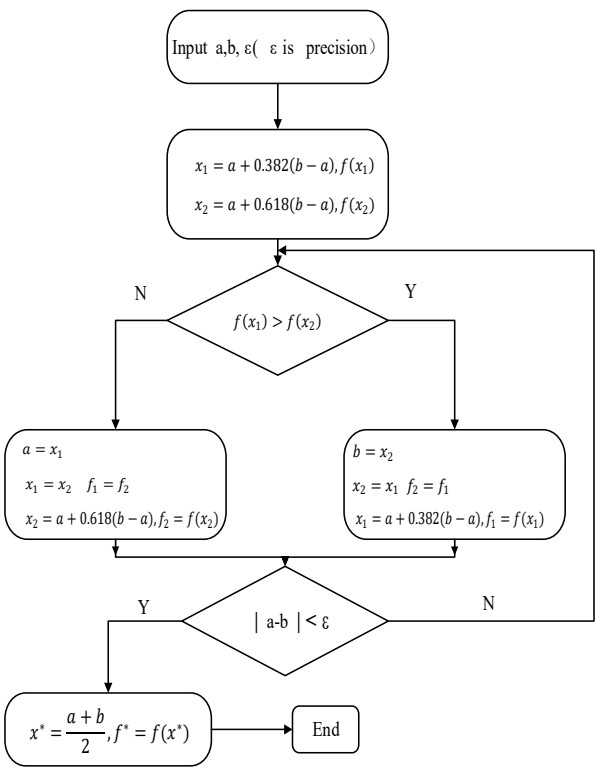

**Figure 9.** Quick search method search process.

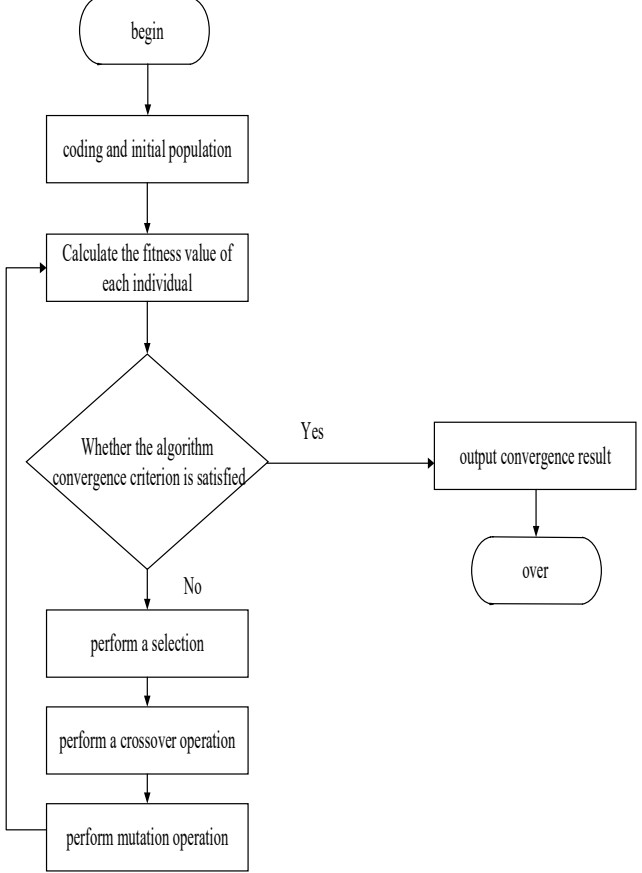

**Figure 10.** Flowchart of the genetic algorithm.

Then on the basis of genetic algorithm, particle swarm optimization (PSO) [65,66], which is widely used in the fields of function optimization, neural network training, and fuzzy control system, is further adopted. The process is shown in Figure 11 [60].

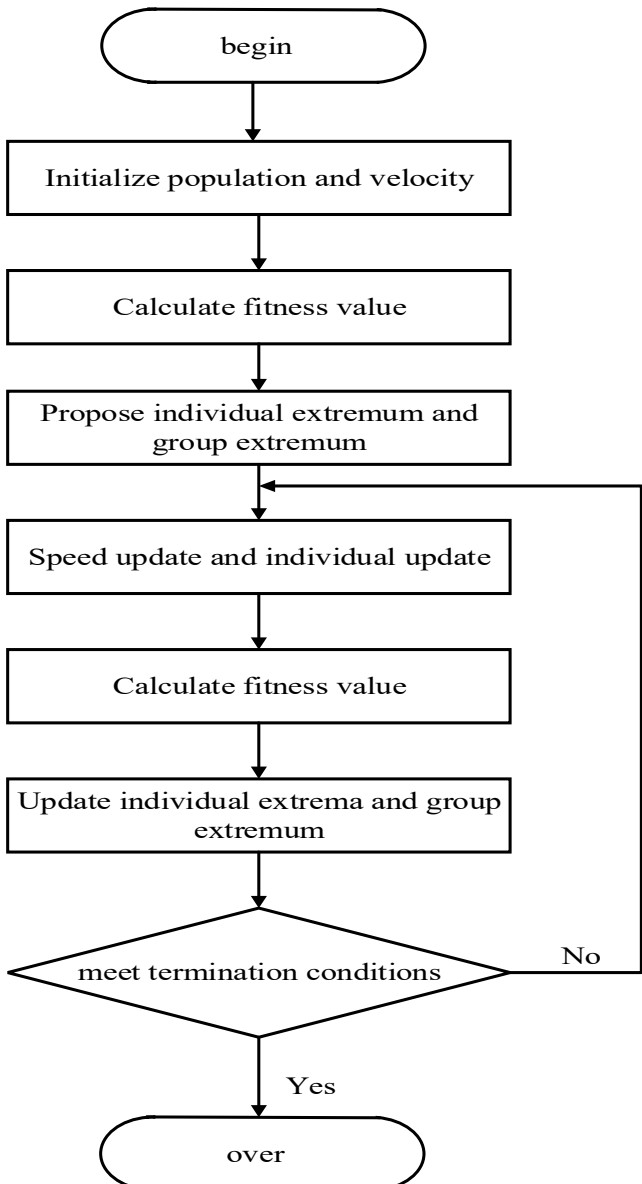

**Figure 11.** Flowchart of standard particle swarm algorithm.

Because the genetic algorithm does not have memory, the offspring will lose the information of the parent during the iteration process, while the particle swarm algorithm is prone to premature convergence and falls into the local optimal solution [67,68]. Therefore, a genetic particle swarm hybrid algorithm (GA-PSO) optimization algorithm based on the particle swarm algorithm and introducing the crossover operation and mutation operation of the genetic algorithm is adopted [69]. The memory function of the particle swarm algorithm for the current optimal solution and the global search function of the genetic algorithm are fully utilized to avoid the optimization algorithm falling into the local optimal solution, and at the same time, the convergence speed is improved and the optimization time is shortened [70]. The process is shown in Figure 12 [60].

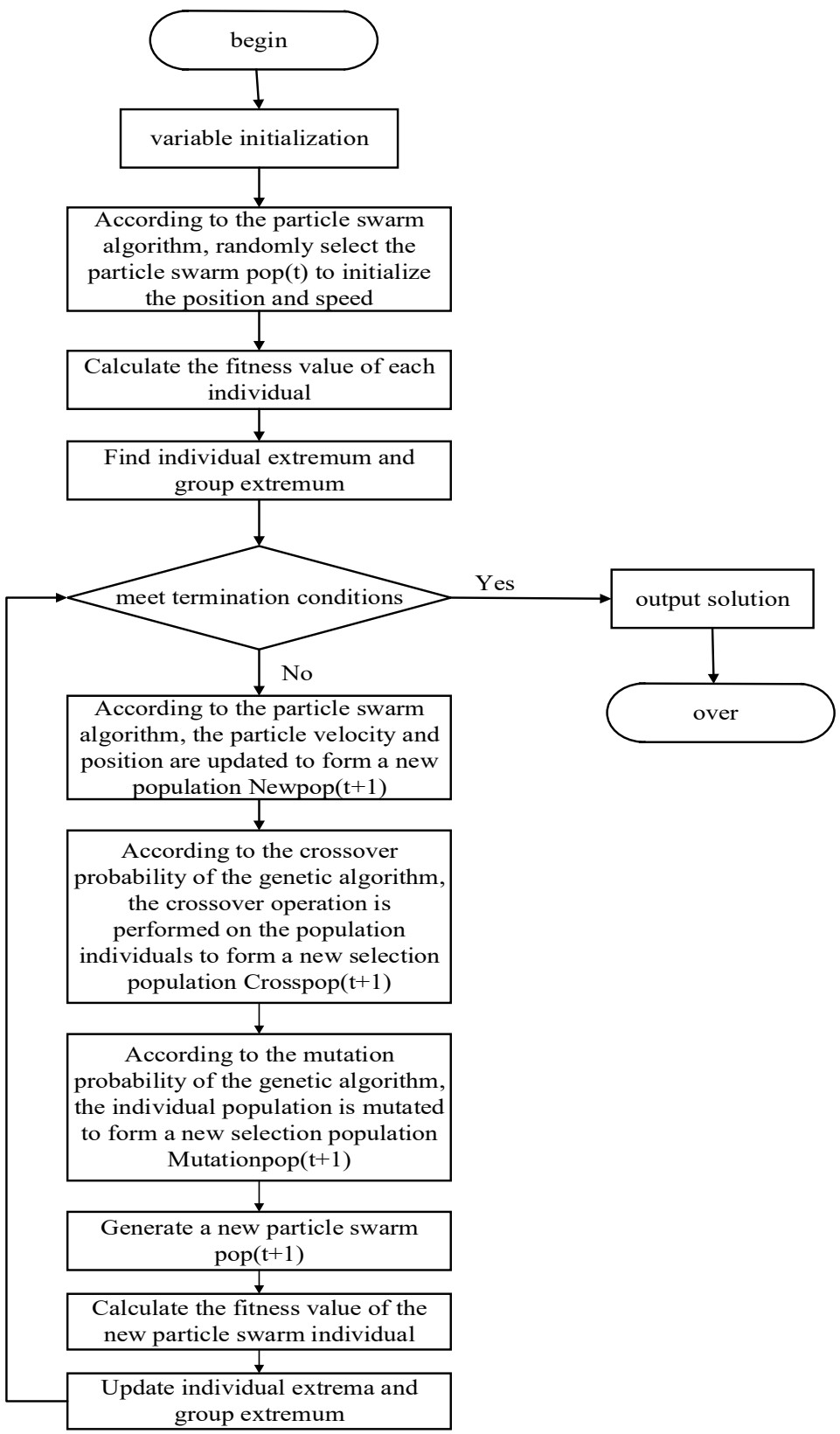

**Figure 12.** Flowchart of the GA-PSO hybrid optimization algorithm.

Finally, the genetic particle swarm optimization algorithm is used to formulate the minimum cornering energy consumption torque distribution coefficient table based on the vehicle dynamics model. The optimal energy-saving contribution of torque distribution

control under different curve conditions is determined, and the online application of torque optimization control is realized. From an economical point of view, the torque distribution standard is formulated by using an optimized algorithm, but there are still some shortcomings in terms of vehicle driving stability and safety. In the future, further research can be carried out in combination with the above ideas and theories.

b.  Torque Distribution for Global Optimization Algorithms

Zheng [71] adopted the dynamic programming control algorithm [72] when considering the global optimization algorithm to transfer the state variables and control variables of the previous moment into the state variables of the current moment, and then, under the constraints of the dynamic system, all the required torque is distributed. Then, the torque distribution strategy under the fuzzy control algorithm [73,74] is analyzed, considering the demand torque $T_{req}$ of the vehicle, so that the driving state of the vehicle is two-wheel or four-wheel drive. When the torque demand of the front wheel drive motor is less than the total torque demand $T_{req}$, the average distribution drive mode is adopted; when the torque demand of the front wheel drive motor is greater than the total torque demand $T_{req}$, a separate two-wheel operation mode is adopted. At the same time, considering the efficiency of the motor, make the motor work in the high-efficiency area.

Based on Model Predictive Control (MPC), Ren [75] designed a Model Predictive Control Energy Saving (MPC-EE) torque distribution algorithm. Using a layered control strategy, the upper controller considers the maximum capacity of motor drive and regenerative braking to determine the required motor torque. The lower layer combines constraint transformation and numerical solution methods to provide an effective method to quickly solve the optimal torque of the motor. In the MPC framework, an objective function with a predicted horizon is designed, and a trial-and-error method is used to determine the horizon value, and the fast numerical algorithm of Ohtsuka (2004) [76] is extended to deal with this kind of inequality. Constrained nonlinear degradation level control is combined with explicit structural and numerical optimization to find the optimal motor torque. The final simulation results also show that the proposed MPC-based strategy can effectively improve energy efficiency and computational efficiency and has better real-time performance. In the future, some conditions such as the saturation constraint of tire slip rate, road conditions, and utilization rate of adhesion coefficient need to be considered for further research.

Based on previous studies, Dizqah [77] believed that the power loss can be reduced by a specific torque distribution algorithm (also known as control distribution (CA) strategy) [78,79]. Based on the reference of the speed of the vehicle, the optimal torque distribution is transformed into a parameter optimization problem. Under the assumption that the power loss of the transmission system increases strictly and monotonically with the torque demand determined by the experiment, the analytical solution of the equal transmission system is given. Based on the power loss characteristics of the power transmission system, the analytical calculation formula of the function of the torque demand threshold and the vehicle speed is proposed, and the simulation and experimental verification results show that the method saves a lot of energy compared with the alternating torque distribution strategy.

### 3.1.3. In-Wheel Motor Drive Torque Distribution Method Considering Both Stability and Energy Efficiency

Zhu [80] first carried out the system dynamics modeling, obtained the dynamic equation of the system, comprehensively considered the system constraints and boundary value conditions, and combined the performance index function considering the motor energy loss and the performance index considering the tire slip energy loss function. The normalization of the two is a performance index function that comprehensively considers the slip energy loss of the motor and the tire, and the torque distribution is constructed as an optimal control problem; the nonlinear state equation and performance function in the optimal control problem are then solved by sequential quadratic programming (SQP) [81]. Further simulation results show that, compared with the torque distribution strategy based

on axle load, the proposed theory can reduce energy consumption and tire wear on the premise of ensuring vehicle stability.

Zhang [82] also adopted the idea of layered control. Based on the premise that the upper layer control ensures the stability of the vehicle during driving under extreme conditions, the lower layer control studies the efficiency of the electric drive system of the vehicle during the driving process. The optimal torque distribution strategy with the optimal and the lowest tire load ratio as the joint control objectives, by taking into account the working characteristics and stability constraints of each driving wheel, optimizes the distribution of torque to each wheel of a distributed drive electric vehicle. The main method under this idea is to carry out multi-objective optimization of the tire load rate of the vehicle and the energy consumption of the electric drive system under the condition of satisfying the generalized force constraints. The optimal torque distribution coefficient under the current operating conditions is solved, and the torque distribution is carried out to achieve the joint control effect of vehicle stability and economy. Firstly, the control objective based on tire load rate is analyzed to ensure the stability of vehicle driving. Then, the control target based on driving energy consumption is analyzed, so as to achieve the effect of efficient energy utilization and energy consumption reduction. Finally, based on the above research, a joint control that takes into account the tire load rate, and the driving energy consumption is designed to ensure vehicle stability and economy at the same time. The particle swarm optimization algorithm (PSO) is used to calculate and solve the straight driving condition and the turning condition, respectively.

De Novellis [83] solved the optimal distribution of torque to a single wheel under specific driving conditions through an optimization-based control distribution (CA) algorithm and proposed an offline optimization design method based on a quasi-static model. They studied the performance of an alternative objective function for optimal wheel torque distribution in a four-wheel drive (4WD) all-electric vehicle. After experimental verification, the results show that the objective function based on the minimum slip criterion has better control performance than the objective function based on the energy efficiency. Energy-based cost functions have marginal benefits in selecting individual wheel torque distributions; in contrast, tire slip distribution-based objective functions allow for smooth changes in wheel torque values under all achievable lateral accelerations and yield only a marginal energy penalty.

Lin [84] proposed a wheel torque distribution strategy based on multi-objective optimization to improve vehicle maneuverability and reduce energy consumption. Due to the model error and parameter error, the upper controller adopts the sliding mode control method to calculate the required yaw moment, and the lower controller adopts the penalty function composed of the yaw moment control bias, the energy loss of the drive system, and the slip constraint [85] to carry out mathematical programming to calculate the wheel torque control distribution. The Newton–Lagrangian algorithm is used to search for the optimal point online from the offline optimization results, and the offline optimization and online optimization are combined for programming to reduce the computational cost. Through MATLAB modeling and simulation, the results show that the energy consumption of the vehicle is reduced when the proposed strategy is used to control the vehicle.

### 3.1.4. Other Distribution Methods of In-Wheel Motor Driving Torque

Wang [86] studied the torque distribution of the eight-wheel hub tram based on the traditional four-wheel hub drive tram. The proposed sub-module idea divides the eight wheels of the vehicle into four parts according to the positional relationship with the center of mass of the vehicle, which are the one- and two-axle wheels on the left, the one- and two-axle wheels on the right, the three- and four-axle wheels on the left, and the three- and four-axle wheels on the right wheel. First, the two wheels of each part are regarded as a whole, a separate mechanical analysis is carried out, a combined analysis is carried out, and then a rule-based torque distribution control strategy is carried out based on the four-quadrant torque distribution priority of friction circle and axle load and compared with the

distribution method based on axle load. Finally, the contribution of each wheel to the yaw moment is compared, and the series chain torque distribution rule is determined, in which the front outer wheel is the main wheel, and the rear four wheels are the auxiliary. The proposed series-chain allocation rule is easy to implement in engineering and has a certain theoretical basis, but the rules can be further adapted according to their own research.

Xu [87] took the human-vehicle-road closed loop as an angle and studied the situation in which the driver controls the vehicle according to the changing traffic conditions and his own driving experience, so that the torque distribution control strategy should be fully integrated with the driver's driving experience and driving intent to improve driving maneuverability and safety. Firstly, the driver's intention is divided into acceleration, braking, and steering manipulation, and the driving intention is distinguished by adaptive reinforcement learning method (AdaBoost) to verify the accuracy of the recognition with a driving simulator. After obtaining the driver's driving intention, a torque distribution strategy based on model predictive control is proposed, the solution problem is transformed into a quadratic programming problem through the cost function, the driving/braking performance of each wheel of the vehicle is described by I/O constraints and road friction conditions, and then the rolling time domain is solved to optimize the control of the vehicle. The process is shown in Figure 13 [87]. The driver's intention model is obtained through the algorithm, the actual constraints in the vehicle driving are considered in the research, and the torque is solved by quadratic programming, which reduces the calculation amount of the controller as much as possible and ensures the calculation speed. However, only in-loop experiments have been carried out, and the next step needs to be combined with the obtained theory to carry out real vehicle experiments to verify.

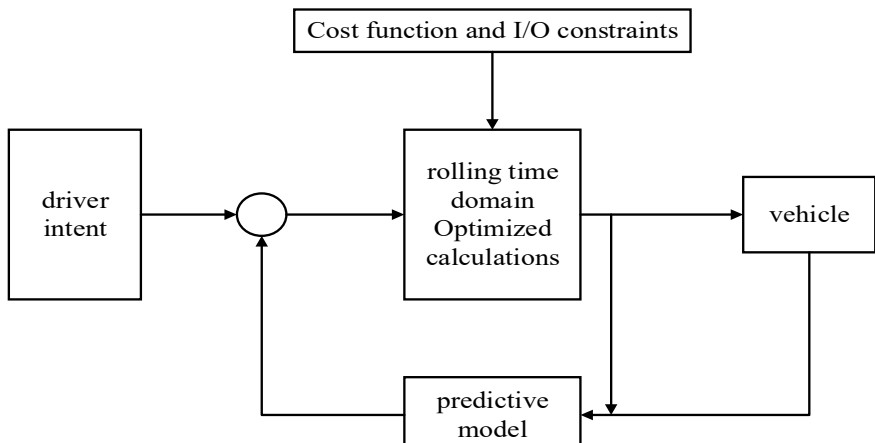

**Figure 13.** Structure diagram of four-wheel independent drive control system.

Chen [88] considered the phenomenon that the instantaneous torque distribution of torque distribution and the actual torque were too large, which led to the instantaneous increase of in-wheel motor current and the increase of energy consumption. The study combined the improvement of motor drive efficiency with the response speed and proposed a torque energy-saving optimization allocation algorithm. The study used MATLAB to calculate the motor efficiency under different torque/speed when given different motor control signals and draw the motor efficiency MAP diagram. Then, combining the respective advantages of the objective function of improving motor efficiency and the objective function of improving motor response speed, an optimal distribution algorithm based on fuzzy controller is proposed, and the optimized torque distribution coefficient is calculated. Then, the cycle condition simulation and real vehicle experiment of vehicle energy consumption are carried out under the three distribution modes of rear axle drive, four-wheel torque average distribution, and torque energy-saving optimal distribution, respectively. The results prove that the proposed torque energy-saving optimal distribution

is good. The strategy can reduce the power consumption of the whole vehicle and achieve the purpose of energy saving.

Yamakawa [89] used the variational principle to minimize the friction work when the tire is in contact with the ground and proposed a method to determine the optimal torque. The friction work done by all tires on the contact area is selected as a function to minimize, and the torque distribution problem is transformed into a function minimization problem to solve, respectively, for slope conditions, cornering conditions, and split conditions. The algorithm is analyzed and verified by simulation.

Kim [90] proposed an adaptive direct yaw moment control method for electric vehicles based on yaw rate model identification. The vehicle yaw rate is solved by PI control, and the torque is distributed by differential braking/drive control. The simulation results show that the proposed method can not only identify the yaw rate model of the electric vehicle, but also utilize the direct yaw moment control. This method controls the lateral motion of an electric vehicle. However, the experimental study will lead to an increase in the side-slip angle of the vehicle's center of mass at high speeds, so further improvement is needed.

Wang [91] calculated the torque load distribution coefficient of each in-wheel motor according to the wheel slip ratio and calculated the torque load coefficient of each in-wheel motor according to the torque load distribution coefficient of each in-wheel motor according to the torque load coefficient of each in-wheel motor. The target driving torque of each wheel is calculated from the operating torque of each in-wheel motor. This solution avoids the problem that the target torque of the whole vehicle is difficult to calculate due to the difference in the working speed of the in-wheel motor and realizes the dynamic joint adjustment between the single wheel to the wheel, to the axle, and then to the torque load factor of the whole vehicle and the torque of the whole vehicle. The transformation of the load scaling factor optimizes the performance of the whole vehicle, makes full use of the output capacity of the front and rear axle motors and batteries in the process of torque distribution, ensures that the capacity of the motors and batteries are maximized without exceeding their allowable limits, and ensures the reliability and stability of the system.

Finally, a comparative analysis of the in-wheel motor torque distribution schemes proposed by the above experts and scholars is carried out, as shown in Table 1.

**Table 1.** In-wheel motor torque distribution scheme and its advantages and disadvantages.

| Torque Distribution Scheme | Advantages | Disadvantage |
| --- | --- | --- |
| A torque vector distribution control method combining differential braking and differential driving [14]. | It can reduce the wheel torque output, improve the utilization rate of the road adhesion limit, and then expand the stability margin of the vehicle. | Without considering the uncertainty of the motor output torque, the robustness of torque distribution needs to be further improved. |
| The driving force distribution based on compound slip rate is proposed on the basis of controlling the driving force distribution based on equal torque and equal power [15]. | While obtaining better longitudinal driving performance, better lateral stability can also be obtained, and the slip ratio of each wheel can be better controlled to tend to an ideal value. | The equal torque and equal power control modes do not fully consider the working state of each wheel, and the compound slip rate control strategy has a large amount of calculation and a long response time when optimizing the current distribution of each wheel on demand. |
| Terminal Sliding Mode Technology (TSMC) [30] | Fast, limited time convergence and high steady-state accuracy allow the driver to more easily steer the vehicle without steering force compensation for the nonlinearity of the vehicle's yaw response. | The fault tolerance integration is not high, and there is jitter interference in the operation process. |

**Table 1.** *Cont.*

| Torque Distribution Scheme | Advantages | Disadvantage |
|---|---|---|
| Based on the error back propagation algorithm and the fastest descent method of BP neural network, the yaw moment algorithm of PID is adjusted [32]. | By considering various constraints, the driving environment of the car can be simulated more realistically, and the adhesion conditions of the road surface can be well used and the life of the motor can be extended when the motor torque is distributed. | The parameters such as the position of the center of mass of the vehicle, the air pressure of the tires and the rolling radius are not considered, and the control target needs to be further refined. |
| Professional drivers conduct experiments with the Adaptive Second-Order Sliding Mode (ASOSM) algorithm [34]. | The research field of optimal torque distribution is broadened. It overcomes the chattering problem of traditional sliding mode and improves the robustness of the control system to model errors and parameter uncert-ainties. | Different drivers' driving styles and driving habits are not considered, and the experimental data are relatively simple, and the information such as geography and external traffic cannot be effectively combined with vehicle dynamics control. |
| Considering factors such as communication delay and interruption in distributed control system, a four-wheel torque optimal distribution algorithm with communication fault tolerance is designed [36]. | The vehicle controller and the motor controller can work in the fault-tolerant mode when there is a network problem, and can switch to the normal mode when the network is normal, which improves the security when the network fails. | Delayed signals in the feedback channel were not considered, nor were long-term delays investigated experimentally. |
| Active Disturbance Rejection Controller (ADRC), quadratic programming algorithm [40]. | It effectively solves the contradiction between overshoot and rapidity of the classical controller, and makes the param-eters of feedback gain and error differential feedback gain easy to set, enhances the robustness of the control system, and improves the accuracy of calculation. | The influence of vehicle controller on the distribution of driving force is not considered, but the effect of vehicle suspension is simplified. |
| Based on the motor efficiency MAP map, a scheme of evenly distributing torque to four wheels is proposed [46]. | For the first time, a new way of studying the torque distribution optimization problem by using the loss analysis method based on the motor efficiency model is proposed. | This scheme has only been tested under specific motor and algorithm conditions, and whether it is applicable to other motor and algorithm conditions remains to be verified by further experiments. |
| A kind of "Energy-and-Adhesion" (EAA) joint optimization to synergistically optimize the energy efficiency of the power system and the utilization rate of road adhesion, with cubic polynomial combined with least squares fitting [50]. | Compared with the quadratic polynomial fitting method and the piecewise linearized fitting method, the proposed cubic polynomial has better effect, and realizes the synergistic optimization of the energy efficiency and steering characteristics of the chassis system under steering conditions. | The calculation steps of the cubic polynomial are cumbersome and take a long time, and the information such as the mass of the vehicle, the position of the center of mass, the vertical load of the wheel, and the side-slip angle of the center of mass of the body are not considered. |
| Fast search method (i.e., golden section method), fuzzy algorithm [56]. | Without being constrained by the parameters of components such as the motor, the driving torque distribution ratio of the front and rear axles can be controlled in the working area where the driving efficiency of the system is optimal. | The algorithm will have a fluctuation problem, which needs to be solved by further calibration, and the fuzzy algorithm will also have a relatively large error. |
| Genetic algorithmparticle swarm optimization, (PSO)GA-PSO [60] | It avoids the optimization algorithm from falling into the local optimal soluteon, improves the convergence speed and shortens the optimization time, determines the optimal energy-saving contribution of torque distribution control under different curve conditions, and realizes the online application of torque optimization control. | The energy consumption of the vehicle in the straight-line driving condition has not been deeply discussed, and the vehicle driving stability and safety are still lacking. |

**Table 1.** *Cont.*

| Torque Distribution Scheme | Advantages | Disadvantage |
|---|---|---|
| Model Predictive Control Energy-Saving (MPC-EE) Torque Distribution Algorithm Based on Model Predictive Control (MPC), Trial and Error Method [75]. | It can effectively improve energy efficiency and computational efficiency, and has better real-time performance. | Some conditions such as tire slip rate saturation constraints, road conditions, adhesion coefficient utilization, etc. need to be further considered. |
| Considering the performance index function of motor and tire slip energy loss comprehensively, the torque distribution is constructed as an opti-mal control problem, and the sequential quadratic programming method (SQP) is used to solve it [80]. | Under the premise of ensuring the stability of the vehicle, it can reduce energy consumption and greatly reduce the wear of tires, and effectively improve the service life of tires. | Only offline simulation verification and analysis are carried out, the solution is time consuming, and the solution efficiency cannot keep up with real-time applications, and the motor characteristics under deceleration braking are not considered. |
| Adaptive reinforcement learning (AdaBoost), cost function [87]. | The trained recognition model can judge the driver's intention more than 98%, and the cost function converts the problem into a quadratic progr-amming problem, which reduces the computational load of the controller as much as possible and ensures the calculation speed. | When considering vehicle dynamics, the effect of suspension system on dynamics is not designed, which brings certain errors to the study of handling stability control. The driver-in-the-loop verification is only carried out under driving simulator conditions. |

References [14,15] can improve the stability of vehicle driving well, but the amount of calculation is large, and the time required is long. While References [30,40] improve the calculation accuracy and speed on the basis of ensuring the driving stability of the vehicle, the experimental design is ideal and does not consider the constraints. References [32,36,56] consider the influence of constraints and are closer to the actual driving situation, but their experimental data are relatively simple and need to be further optimized. References [46,50] constrain vehicle energy efficiency in novel perspectives, but both require experiments under specific motors and experimental conditions. References [56,60] can improve the energy utilization efficiency without the constraints of the motor, but do not consider the stability of the vehicle. References [80,87] comprehensively consider the stability and energy efficiency of the vehicle, so that the stability and energy efficiency of the vehicle are improved.

## 4. Conclusions

(1) Experts and scholars at home and abroad have established various models according to their own research purposes, such as the motor control signal generator model, which can accurately control the motor signal. The torque distribution model with the driving efficiency of the whole vehicle as the control target can comprehensively consider the vehicle and road conditions, so that the experiment is close to the actual working conditions. The vehicle dynamics model adopts the idea of modular modeling, and comprehensively considers various modules, so that the model structure is complete and clear. The vector control and simulation model is established from the perspective of vector control, which can avoid some noise interference and better control the motor. The motor efficiency model, starting from the motor loss, studies the motor efficiency under the driving condition and the braking condition, respectively, so that the experimental results are closer to the real value. The commonly used 8-DOF and 13-DOF vehicle dynamics models also have a high degree of matching for subsequent experimental studies.

(2) In the torque distribution scheme, the torque vector distribution control method combining differential braking and differential drive, and the driving force distribution method based on compound slip rate, can better improve the utilization rate of road adhesion limit, the control program is relatively easy to implement, and the calculation method is also simple and easy to implement. The proportional torque distribution and the optimal distribution method based on the weight of the center of mass slip angle and the hierarchical torque distribution strategy under emergency conditions fully consider

the tire force factor, improve the stability margin of the in-wheel motor-driven electric vehicle, and improve the driving stability of the vehicle. The terminal sliding mode technology (TSMC) is used to track the desired vehicle motion by interpreting the driver's command, and from the driver's point of view and asking professional drivers to conduct experiments, the expected torque of the professional driver is obtained, taking into account the lateral and longitudinal stability. To carry out the optimal torque distribution, from the perspective of the influence of the driver's operation on the driving of the car, the algorithm is used to optimize the driver's behavior, thereby improving the stability and energy consumption of the vehicle. The four-wheel torque optimal distribution algorithm with communication fault tolerance can still work well when the vehicle controller and motor controller have network problems, avoiding the danger caused by network failures. The fast search method (that is, the golden section method) and the model predictive control energy saving (MPC-EE) torque distribution algorithm based on model predictive control (MPC) can effectively improve energy efficiency and computational efficiency and have good real-time performance.

(3) Future research should use modern and rapidly developing science and technology and use multi-disciplinary and multi-field technology to develop experimental research on in-wheel motor torque distribution. It cannot only stay in the stage of offline simulation verification. How to combine relevant theoretical research results and realize real vehicle test verification under complex and diverse working conditions is one of the important research directions in the future. In future research, it is necessary to comprehensively consider various constraints in actual driving and to simulate the experimental environment more realistically, so that the designed torque distribution scheme is more practical. The current torque distribution scheme only conducts unilateral research on safety or energy efficiency. There are few in-wheel motor driving torque distribution schemes that comprehensively consider safety and energy efficiency. The torque distribution scheme designed in the future needs to strengthen the safety and comprehensive research on energy efficiency.

**Author Contributions:** S.H. formulated the overall research objectives, carried out the research design, and manuscript writing. X.F. and Q.W. formulated the overall research objectives. X.C. and S.Z. made contributions to the research idea. All authors have read and agreed to the published version of the manuscript.

**Funding:** This research was supported by the Fundamental Research Funds for the Universities of Henan Province, grant number NSFRF220435, Key Scientific and Technological Project of Henan Province, grant number 222102220024, 212102210050, Collaborative Education Project of Industry and University Cooperation of the Ministry of Education, grant number 202102449067.

**Institutional Review Board Statement:** Not applicable.

**Informed Consent Statement:** Not applicable.

**Data Availability Statement:** All data in this study are available in the documents referenced in bibliography.

**Conflicts of Interest:** This article does not have any conflict of interest, it was completed by myself, and the manuscript was approved by all authors for publication.

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
