# Peer review of "Review on Torque Distribution Scheme of Four-Wheel In-Wheel Motor Electric Vehicle"

_machines, doi:10.3390/machines10080619_

Round 1

Reviewer 1 Report

The paper is interesting but difficult to read and easy assimilation of knowledge. In my opinion it is becouse the paper is to wide and presented knowledge has not fully been thought out and ordered. Used style presented next authors and shortly their work. In my opinion it should be ordered to presents some areas of knoledge connected with the topic. It should be ordered in a that way that firstly the problem will be shown and then technical solution of them with some necessary explanation. 

Maybe some aspects could be shorten to give the place for the main topic. 

For example: 3.1 In-wheel motor and vehicle dynamics modeling - in my opinion is not necessary, especially in the form of other authors work review. There is also a question if ale the formulas (1) to (5) are necessary to understad next parts of the paper? 

Some parts presenting the basic knoledge form vehicle dynamics fundamentsla are not neccesary in that type of work. 

It seems that paper could start from "3.2 In-wheel motor drive torque distribution method".

There are such subtitles in the paper:

3.2.1 In-wheel motor drive torque distribution method to improve stability

(1) In-wheel motor drive torque distribution method based on vehicle dynamics model to improve stability:

a. Torque distribution considering road adhesion

b. Torque distribution considering tire force factors

c. Torque distribution considering driver factors

and for the point:

(2) In-wheel motor drive torque distribution method based on algorithm to improve stability - very similar to previous. 

a. Calculate the torque allocation of the optimal yaw moment algorithm

b. Torque distribution of quadratic programming algorithms

a. Torque distribution considering motor energy efficiency optimization

3.2.2 In-wheel motor drive torque distribution method to improve energy efficiency

(1) In-wheel motor drive torque distribution method based on vehicle dynamics model to improve energy efficiency 

a. Torque distribution considering motor energy efficiency optimizationIn my oponion the most important part is presentation of different goals, ideas and later comparision of ideas and technical solutions.

b. Torque distribution considering tire slip energy loss 

(2) In-wheel motor drive torque distribution method based on algorithm to improve energy efficiency 

a. Torque Distribution of Multi-motor Cooperative Control Algorithm

b. Torque Distribution for Global Optimization Algorithms

Some diagram for different tested or presented solutions  should be made to order all acquired knowledge. 

Some small comments:

Line 89, Figure 1.

Figure 1. Some description in the figure would make it easier to understand.

Line 92

The in-wheel motor is a structure that directly integrates the motor with the driven  tire – tire or a wheel?

Line 143

Diagram produces a question – there is a in-wheel motor on the left and right side of a figure. How to interpret it?

Line 179 – it should be explained why 8 DOF – it should be added that there is a 4 wheels..

Lines from 215 to 258 presents vehicle dynamics basic theory or electric motor basic theory. Is it necessary for the topic of the paper to present it here…?

Line 291

“…differential 290 braking and differential braking were adopted”…not braking and driving?

Line 331. Figure 8. Frame design for vehicle yaw moment control – it is not able to fully understand this control only from presented figures.

 Line 622

Figure is difficult to read – the aspect ratio is not natural

The idea of the paper is good, but it could be better written (easier to understand)

Author Response

Dear Reviewer,

Thank you very much for your valuable suggestions for my manuscript entitled " Review on Torque Distribution Scheme of Four-wheel In-wheel Motor Electric Vehicle". Now I will respond to your question in the following content.

1.For example: 3.1 In-wheel motor and vehicle dynamics modeling - in my opinion is not necessary, especially in the form of other authors work review. 

Answer: Respecting the reviewer's opinion, the content of section 3.1 is not helpful to the work of this paper, so it has been deleted.

2.Line 89, Figure 1.

Figure 1. Some description in the figure would make it easier to understand.

Answer: Some structures in Figure 1 are marked.

3.Line 92

The in-wheel motor is a structure that directly integrates the motor with the driven  tire – tire or a wheel?

Answer:Line 92, the hub motor is a structure that directly integrates the motor with the wheel, which is modified and marked in the text.

4.Line 143

Diagram produces a question – there is a in-wheel motor on the left and right side of a figure. How to interpret it?

Answer: In line 143, the in-wheel motor on the right side of the figure distributes the torque calculated by the torque distribution algorithm to the in-wheel motor, and then feeds it back to the car for prediction and calculation of the next step.

5.Line 179 – it should be explained why 8 DOF – it should be added that there is a 4 wheels.

Answer: Section 3.1 has been deleted in respect of the reviewer's opinion, and the content of line 179 about 8 degrees of freedom has been deleted.

6.Lines from 215 to 258 presents vehicle dynamics basic theory or electric motor basic theory. Is it necessary for the topic of the paper to present it here…?

Answer: Section 3.1 has been deleted in respect of the reviewer's comments, and the contents of lines 215 to 258 have been deleted.

7.Line 291

“…differential 290 braking and differential braking were adopted”…not braking and driving?

Answer: Line 291, with differential drive and differential braking, modified and marked in line 149 in the text.

8.Line 331. Figure 8. Frame design for vehicle yaw moment control – it is not able to fully understand this control only from presented figures.

Answer: Line 331, Figure 8 is the overall structural block diagram. By obtaining the ideal values of the front wheel angle and vehicle speed, the total motion force and total torque are calculated through the sliding mode surface switching mechanism of the upper controller, and then the lower controller rotates in opposite directions. The moments are distributed and fed back to the vehicle model. The overall steps are explained below.

9.Line 622

Figure is difficult to read – the aspect ratio is not natural

Answer: Line 622, the image is resized.

Reviewer 2 Report

The paper presents an analysis of torque distribution scheme for four-wheel in-wheel motor electric vehicle. The paper is well written. 

Author Response

Dear Reviewer,

Thank you very much for your valuable suggestions for my manuscript entitled " Review on Torque Distribution Scheme of Four-wheel In-wheel Motor Electric Vehicle". Now I will respond to your question in the following content.

Article revised and improved for English language and style

This manuscript is a resubmission of an earlier submission. The following is a list of the peer review reports and author responses from that submission.

Round 1

Reviewer 1 Report

The authors of the paper attempt to make a review on torque distribution schemes of four-in-wheel motor electric vehicles (EVs). It is an important topic.

In general, the most important issues of the drive configuration are the drivability (performance) and vehicle safety, follow by energy efficiency, stability, and comfort. However, the paper focuses on torque distribution’s review with regards to energy efficiency and stability without reasoning, that might make readers raising questions

Other details comments are as follows:

1.       There are two sections with number 1 (Introduction and Classification).

2.       Section “1.1Driving forms of traditional automobiles” is not necessary at all in this paper, because subject has been already clear from the title: “Four-wheel In-wheel Motor Electric Vehicle”.

3.       Many figures need reference, for example: Fig. 2, 3, 4, 5, etc., especially the papers copied from internet (e.g. Fig. 4)

4.       Figures 6, 7, 9, 10, 11, 12, 13, 14, 15, 16, 17, 18 should have clearly reference

5.       Section “2.1 In-wheel motor and vehicle dynamics modeling”

a.       The mathematical expressions (equations 1 - 4) are too few, which can not give a good modeling of the four-in-wheel motor EVs.

b.       In addition, Fig. 8 is the simplified “bicycle-like” model, which is not much help for the study of “Four-wheel In-wheel Motor Electric Vehicle”.

c.       Moreover, there is no description (equation) on the energy efficiency, stability, which is the focus of the paper.

6.       Section “2. In-wheel motor drive torque distribution scheme”: the authors present several schemes. However, their classification into sub-sections is not so representative for a huge amount of works in literature.

7.       Table 2 reports some schemes, with advantages and disadvantages

a.       The reference of each schema should be provided

b.       The comparison between these schemas should be carried out.

8.       Reference: there are 105 references. However, there is a serious imbalance domestic/international works. In-wheel Motor EV is not a new research topic. It has been studied for more than 3 decades. There are several strong research Labs around the world. The list of reference is not so representative among thousand papers in literature.

Reviewer 2 Report

- Abstract is long and unclear. The abstract must be concise and with the idea. 

- Introduction must give the story of why this review paper is important!

- Also, in the Introduction note paper organization.

- The table is in the high font. this Tables must be organized in a readable manner.

- Text in Fig 1, 6, 7, 9, 12 is too high and different.

- Fig. 2 is too simple and very well known

- Fig. 3 is not quality

- Correct Fig 14. A flowchart is not ok.

- Conclusion must be the conclusion - not main paper goals in bullet style.